# Cervical cancer screening utilization and predictors among eligible women in Ethiopia: A systematic review and meta-analysis

**Melaku Desta**[1]*, **Temesgen Getaneh**[1], **Bewuket Yeserah**[1], **Yichalem Worku**[1], **Tewodros Eshete**[2], **Molla Yigzaw Birhanu**[3], **Getachew Mullu Kassa**[1], **Fentahun Adane**[4], **Yordanos Gizachew Yeshitila**[5]

1 Department of Midwifery, College of Health Sciences, Debre Markos University, Debre Markos, Ethiopia, 2 Department of Public Health, College of Health Sciences, Debre Markos University, Debre Markos, Ethiopia, 3 College of Health Science, Debre Markos University, Debre Markos, Ethiopia, 4 Department of Biomedical Sciences, School of Medicine, Debre Markos University, Debre Markos, Ethiopia, 5 Departments of Nursing, School of Health Sciences, Arba Minch University, Arba Minch, Ethiopia

* melakd2018@gmail.com

## Abstract

### Background

Despite a remarkable progress in the reduction of global rate of maternal mortality, cervical cancer has been identified as the leading cause of maternal morbidity and mortality, particularly in sub-Saharan African countries. The uptake of cervical cancer screening service has been consistently shown to be effective in reducing the incidence rate and mortality from cervical cancer. Despite this, there are limited studies in Ethiopia that were conducted to assess the uptake of cervical cancer screening and its predictors, and these studies showed inconsistent and inconclusive findings. Therefore, this systematic review and meta-analysis was conducted to estimate the pooled cervical cancer screening utilization and its predictors among eligible women in Ethiopia.

### Methods and findings

Databases like PubMed, Web of Science, SCOPUS, CINAHL, Psychinfo, Google Scholar, Science Direct, and the Cochrane Library were systematically searched. All observational studies reporting cervical cancer screening utilization and/ or its predictors in Ethiopia were included. Two authors independently extracted all necessary data using a standardized data extraction format. Quality assessment criteria for prevalence studies were adapted from the Newcastle Ottawa quality assessment scale. The Cochrane Q test statistics and $I^2$ test were used to assess the heterogeneity of studies. A random effects model of analysis was used to estimate the pooled prevalence of cervical cancer screening utilization and factors associated with it with the 95% confidence intervals (CIs). From 850 potentially relevant articles, twenty-five studies with a total of 18,067 eligible women were included in this study. The pooled national cervical cancer screening utilization was 14.79% (95% CI: 11.75, 17.83). The highest utilization of cervical cancer screening (18.59%) was observed in Southern Nations Nationalities and Peoples' region (SNNPR), and lowest was in Amhara region

**Data Availability Statement:** All relevant data are within the manuscript and its Supporting Information files.

**Funding:** The author(s) received no specific funding for this work.

**Competing interests:** The authors have declared that no competing interests exist.

**Abbreviations:** CBCS, Community Based Cross-sectional Studies; FBCS, Facility- Based Cross-sectional Studies; HIV, Human Immunodeficiency Virus; HPV, Human Papilloma Virus; LMICs, Low and Middle-Income Countries; POR, Pooled Odds Ratio; SNNPR, Southern Nations, Nationalities and Peoples Representative; SSA, Sub Saharan Africa; STI, Sexually Transmitted Infection, WHO: World Health Organization.

(13.62%). The sub-group analysis showed that the pooled cervical cancer screening was highest among HIV positive women (20.71%). This meta-analysis also showed that absence of women's formal education reduces cervical cancer screening utilization by 67% [POR = 0.33, 95% CI: 0.23, 0.46]. Women who had good knowledge towards cervical screening [POR = 3.01, 95%CI: 2.2.6, 4.00], perceived susceptibility to cervical cancer [POR = 4.9, 95% CI: 3.67, 6.54], severity to cervical cancer [POR = 6.57, 95% CI: 3.99, 10.8] and those with a history of sexually transmitted infections (STIs) [POR = 5.39, 95% CI: 1.41, 20.58] were more likely to utilize cervical cancer screening. Additionally, the major barriers of cervical cancer screening utilization were considering oneself as healthy (48.97%) and lack of information on cervical cancer screening (34.34%).

## Conclusions

This meta-analysis found that the percentage of cervical cancer screening among eligible women was much lower than the WHO recommendations. Only one in every seven women utilized cervical cancer screening in Ethiopia. There were significant variations in the cervical cancer screening based on geographical regions and characteristics of women. Educational status, knowledge towards cervical cancer screening, perceived susceptibility and severity to cervical cancer and history of STIs significantly increased the uptake of screening practice. Therefore, women empowerment, improving knowledge towards cervical cancer screening, enhancing perceived susceptibility and severity to cancer and identifying previous history of women are essential strategies to improve cervical cancer screening practice.

## Background

Despite a remarkable progress in the reduction of maternal mortality, cervical cancer is the second most commonly diagnosed cancer and the leading cause of cancer related death among African women [1]. There were approximately 236,000 deaths from cervical cancer worldwide and it was the most common cancer in east and middle Africa [2, 3]. About 90% of cases and 85% of these deaths have occurred in Low and Middle-Income Countries (LMICs); the highest has occurred in Sub-Saharan Africa (SSA) and approximately 311,000 women died from cervical cancer [2]. The incidence, the death rate and morbidities associated with cervical cancer significantly varies across the world; higher in the developing nations compared to the developed countries [4]. The high burden of cervical cancer is mainly due to the early onset of sexual intercourse, multiple sexual partners, human immunodeficiency virus (HIV) infection, history of sexually transmitted infections (STIs), human papilloma virus (HPV) infection, cigarette smoking, limited resources for early detection and poor HPV vaccination coverage [5, 6].

Almost all of the maternal deaths associated with cervical cancer could be prevented if early and effective interventions mechanisms to cervical cancer control were available to all women. In particular, a comprehensive approach such as prevention, early diagnosis, effective screening and treatment programmes of pre-cervical lesions are essential for prevention of cervical cancer [7]. Visual inspection with Acetic Acid (VIA) and Visual Inspection with Lugol's Iodine (VILI) are commonly used in low-resource settings [6]. VIA combined with the immediate treatment of women who tested positive at the first visit was cost saving and was the next most effective strategy, with a 26% decrease in the incidence of CC, further reduce mortality due to

CC. A large-cluster randomized trial from rural India showed that a single round of HPV screening could reduce the incidence and mortality from CC of approximately 50% [8].

The guidelines of the World Health Organization (WHO), the United States Preventive Services Task Force (USPSTF) and the American Cancer Society (ACS) recommends that all eligible women should have cervical cancer screening at least once every three years [9]. Ethiopia adopted WHO's recommendation that woman aged 30 and above should begin screening for cervical cancer at least one to three years of age with a see- and -treat approach. However, sexually active and HIV-positive women (start screening at HIV diagnosis) are suggested to be screened every 3 years regardless of their age [10]. The prevalence of cervical cancer screening is much higher at the Western countries than SSA [11, 12]; 85.0% in the United States, 78.6% in the United Kingdom [13], and ranges from 2% in Ethiopia, 6% in Kenya [14], to 8% in Nigeria [15]. The lower rate of cervical cancer screening programme at LMICs may be related to the complexity of the screening process and the common inherent barriers in the setting such as poverty, limited access to information, lack of knowledge of cervical cancer, lack of healthcare infrastructure required, lack of trained practitioners and the absence of sustained prevention programmes [16].

The government of Ethiopia launched a cervical cancer screening service and has given more emphasis on programs focusing on the early detection of cervical cancer using advocacy efforts by different stakeholders such as academia, professionals, media and partners. However, the prevalence of cervical cancer remains a major problem, and it is one of the leading causes of morbidity and mortality among women in the country [17, 18]. Evidence show success of cervical screening initiatives depend on high participation of the target population, which in turn is determined by the women's knowledge, perceptions, health orientations and other socio-cultural issues. It is also affected by factors including early marriage, early sexual practice, delivery of the first baby before the age of 20, multiple sexual partners and low socio economic status. Therefore, addressing the different barriers for poor utilization of cervical cancer screening is essential component of intervention. Although, there were previous pocket studies conducted on these issues in Ethiopia, the studies showed fragmented, inconsistent and inconclusive findings. Even the studies were fragmented in different specific population characteristics like among HIV positive women and reproductive age women. Therefore, this systematic review and meta-analysis aimed to estimate the pooled cervical cancer screening utilization and its predictors among all eligible women in Ethiopia. It also aimed to address the common barriers of cervical cancer screening.

## Methods

### Registration of systematic review, data sources and search strategies

The purpose of this systematic review and meta-analysis was to estimate the pooled utilization level of cervical cancer screening and its predictors among women of reproductive-aged in Ethiopia. The protocol has been registered with the International Prospective Register of Systematic Review (PROSPERO), the University of York Center for Reviews and Dissemination (https://www.crd.york.ac.uk/), registration number **CRD42019119626**. The findings of this review have been reported as recommended by the Preferred Reporting Items for Systematic Review and Meta-Analysis (PRISMA-P) 2009 statement checklist [19] (S1 Table). All published articles were searched from major international databases like PubMed, Cochrane Library, Psych Info, Scopus, CINAHL, Web of Science, Science Direct, Google Scholar and African Journals Online. Additionally, Google hand searches were used mainly for unpublished studies. A search was also made for the reference list of studies already identified in

order to retrieve additional articles. The Population, Exposure, Comparison and Outcomes (PECO) search formula was used to retrieve articles.

All eligible women for cervical cancer screening Ethiopia were the population of interest for this study. The outcome of interest was the utilization of cervical cancer screening among women. The predictor variables of cervical cancer screening utilization included in this study were age of women, educational status, and occupational status, knowledge of cervical cancer screening, perceived susceptibility and severity to cervical cancer and history of sexually transmitted infections. Comparisons were defined for each predictor based on the reported reference group for each predictor in each respective variable.

For each of the selected components of PECO, electronic databases were searched using the keyword search and the medical subject heading [MeSH] words. The keywords include "utilization, uptake, cervical cancer, screening, and women of reproductive age as well as Ethiopia". The search terms were combined by the Boolean operators "OR" and "AND. The specific searching detail in PubMed was putted in S1 Appendix.

## Eligibility criteria and study selection

This review included studies that reported either the use of cervical cancer screening or the cervical cancer screening predictors in Ethiopia. All published and unpublished studies through April 7, 2020 and reported in English language were retrieved to assess eligibility for inclusion in this review. However, this review excluded studies that were case reports of populations, surveillance data (demographic health survey), and abstracts of conferences, articles without full access and the outcome of interest not reported. The article selection underwent several steps. Two reviewers (MD and TE) evaluated the retrieved articles for inclusion using their title, abstract and full text review. Any disagreement during the selection process between the reviewers was resolved by consensus. Full texts of selected articles were then evaluated using the *prior* eligibility. During the encounter of duplication; only the full-text article was retained.

## Quality assessment and data collection

The Newcastle-Ottawa Scale (NOS) quality assessment tool was used to assess the quality of the included studies. The tool contains three components- selection of the study groups, comparability of the study groups, and ascertainment of exposure or outcome [20]. The main component of the tool was graded from five stars and mainly emphasized on the methodological quality of each primary study. The other component of the tool graded from two stars and mainly concerned with the comparability of each study. The last component of the tool was graded from three stars and was used to evaluate the results and statistical analysis of each original study. The NOS included three categorical criteria with a maximum score of 9 points. The quality of each study was assessed using the following score algorithms: ≥7 points were considered as "good", 4 to 6 points were considered as "moderate", and ≤ 3 point was considered as "poor" quality studies. In order to improve the validity of this systematic review result, only primary studies of fair to good quality have been included. The two reviewers (MD and TE) independently assessed articles for overall study quality and extracted data using a standardized data extraction format. The data extraction format included primary author, year of publication, region of the study, sample size, prevalence, and the selected predictors of cervical cancer screening utilization.

## Publication bias and statistical analysis

The publication bias was assessed using the Egger's [21] and Begg's [22] tests with a p-value of less than 0.05. The $I^2$ statistic was used to assess heterogeneity between studies and a *p-value* of

less than 0.05 was used to detect heterogeneity. As a result of the presence of heterogeneity, a random-effects model was used as a method of analysis [23]. Data were extracted in Microsoft Excel and exported to Stata version 11 for analysis. Subgroup analysis was conducted by geographic region, population's characteristics and design or type of study. Moreover, a meta-regression model based on sample size and year of publication was used to identify the sources of random variations in the included studies. The effect of selected determinant variables was analyzed using separate categories of meta-analysis [24]. The findings of the meta-analysis were presented using forest plots and Odds Ratio (OR) with its 95% Confidence intervals (CI). In addition, we conducted a sensitivity analysis to assess whether the pooled prevalence estimates were influenced by individual studies.

## Results

### Study identification and characteristics of included studies

This systematic review and meta-analysis included both published and unpublished studies on the use of cervical cancer screening in Ethiopia. A total of 850 articles were found from the review. Of these, 250 duplicated records were removed and 581 articles were excluded by screening using their titles and abstracts. Subsequently, a total of 38 full-text papers were assessed for eligibility on the basis of the inclusion and exclusion criteria. Thus, four studies were excluded due to lack of the outcome of interest [25–30], three due to low quality [31–33], five due to difference in the study population [34–39] and only one study was excluded due to lack of access to the full text [40]. Finally, 25 studies were included in the final quantitative meta-analysis (Fig 1).

All of the included studies were cross-sectional. From this, twelve studies were facility-based cross sectional studies (FBCS) and thirteen were community- based cross-sectional studies (CBCS). The review was conducted among 18,067 women to estimate the pooled prevalence of cervical cancer screening. Publication of articles was between 2016 and 2020. The largest sample size was 5,823 women in a national level study [41] and the smallest sample was 250 women from a study conducted in Oromia region [42]. All studies were conducted in five geographic regions of Ethiopia. Four studies (16%) were from Addis Ababa [43–46], nine (36%) were from Amhara [47–55], four (16%) were from Southern Nations, Nationalities and Peoples Representative (SNNPR) [56–60], four (16%) were from Oromia [42, 61–63], two (8%) were from Tigray [64, 65], and the remaining one study [41] was a national- level study. Twelve studies were conducted among eligible women with no specific characteristics of their HIV status [44, 47], five studies on HIV-positive women [43, 48, 53, 61, 63], four studies among healthcare workers [59, 63, 65, 66] and the remaining one study [51] was conducted among women who were commercial sex workers (Table 1).

### Meta-analysis of cervical cancer screening utilization in Ethiopia

The highest cervical cancer screening utilization was observed in SNNPR, a study conducted at ART health facilities of Hawassa, 40% [57] and Wolayita hospitals, 22.9% [60]. Whereas, the lowest was 2.9% in a national level study [41] and 5.4% from a study conducted in Amhara region [54].

The meta-analysis of twenty-five studies showed that the pooled national level of cervical cancer screening utilization was 14.79% (95% CI: 11.75, 17.83). A random-effect model of analysis was used due to significant heterogeneity ($I^2$ = 97.9%, *p-value*<0.05) (Fig 2). Publication bias was assessed using Eggers test and it was statistically significant, *p-value* less than 0.0001. To account for publication bias, the duval and trimmed full analysis was performed. The univariate meta-regression model was also used to identify possible sources of heterogeneity using different covariates like year of publication and sample size. However, none of these

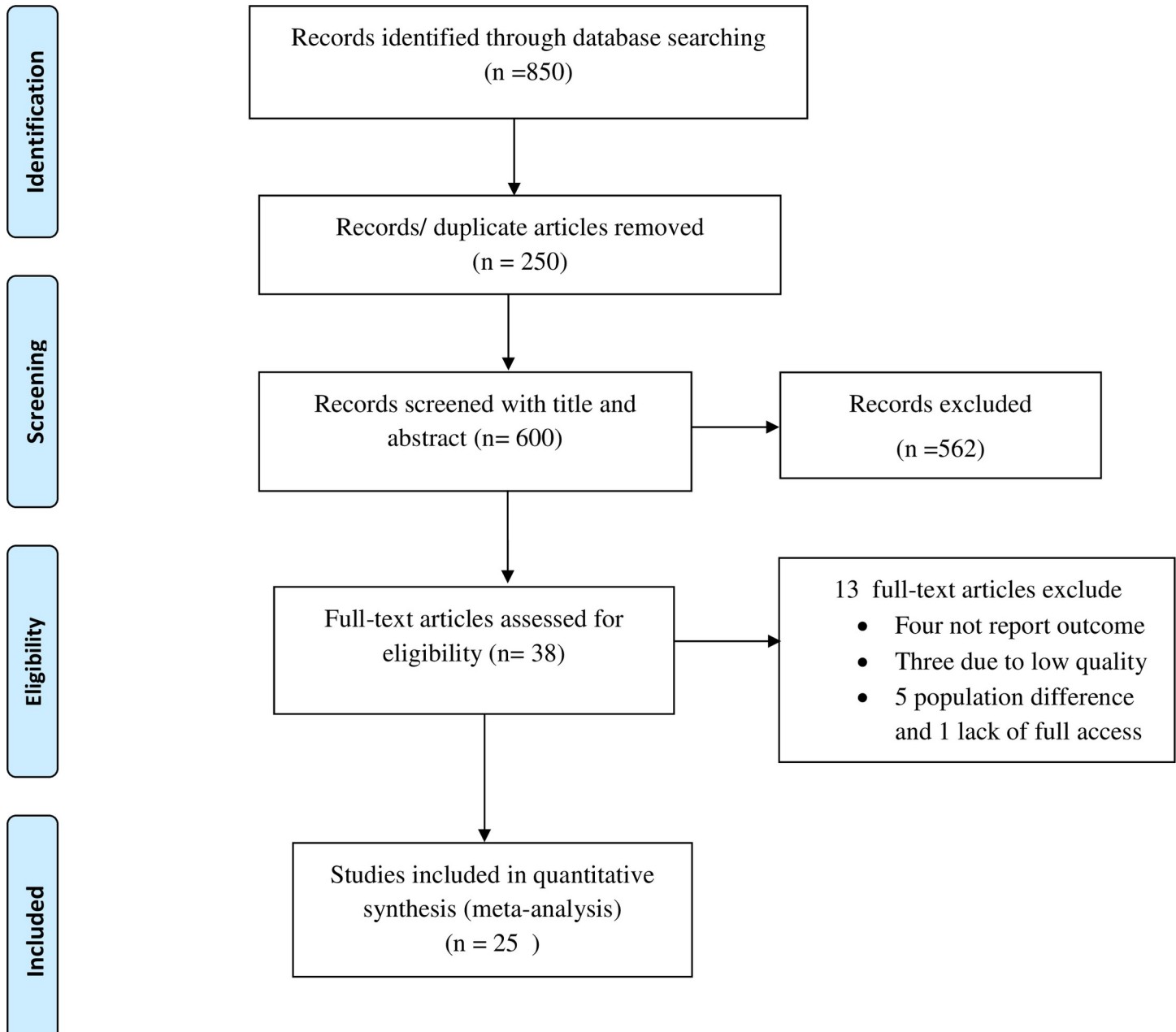

**Fig 1. PRISMA flow diagram of cervical cancer screening utilization in Ethiopia.**

variables were found to be statistically significant, *p-value* > 0.05. Moreover, the sensitivity analysis using a random-effects model showed that no single study had unduly influenced the overall estimate of the use of cervical cancer screening among Ethiopian women (S1 Fig). The funnel plot also showed that there was symmetrical distribution (Fig 3).

## Subgroup analysis

The subgroup analysis was conducted based on region of studies, the study design and women's characteristics. Therefore, this random effect meta-analysis based on the geographic

**Table 1. Characteristics of the included studies in the meta-analysis, Ethiopia.**

| Author | Year | Region | Prevalence | Design | Sample | Population |
|---|---|---|---|---|---|---|
| Shiferaw H et al. [43] | 2018 | AA | 10.8 | FBCS | 598 | HIV+ |
| Getachew S et al. [44] | 2018 | AA | 25 | FBCS | 520 | All |
| Bante SA et al. [47] | 2019 | Amhara | 20.9 | CBCS | 577 | All |
| Aweke YH et al. [56] | 2017 | SNNPR | 9.9 | CBCS | 583 | all |
| Nega AD et al. [48] | 2018 | Amhara | 10 | FBCS | 496 | HIV+ |
| Nigussie T et al. [49] | 2019 | Amhara | 15.5 | CBCS | 737 | all |
| Bayu H et al. [64] | 2016 | Tigray | 19.8 | CBCS | 1186 | all |
| Assefa AA et al. [57] | 2019 | SNNPR | 40.1 | FBCS | 342 | all |
| Gebreegziabher M et al. [65] | 2016 | Tigray | 10.7 | FBCS | 225 | all |
| Solomon K et al. [61] | 2019 | Oromia | 25 | FBCS | 475 | HIV+ |
| Tefera and Mitiku [50] | 2017 | Amhara | 11 | CBCS | 620 | All |
| Muluneh BA et al. [51] | 2019 | Amhara | 13.28 | CBCS | 467 | CSWs |
| Seyoum T et al. [58] | 2017 | SNNPR | 9.6 | FBCS | 281 | all |
| Geremew AB et al. [26] | 2018 | Amhara | no data | 1152 | 98.7 | |
| Michael E et al. [42] | Unpub | Oromia | 17.6 | CBCS | 250 | all |
| Galibo T et al. [41] | 2017 | National | 2.9 | CBCS | 5823 | all |
| Kassa AS et al. [52] | 2018 | Amhara | 7.3 | CBCS | 735 | all |
| Erku DA et al. [53] | 2017 | Amhara | 23.5 | FBCS | 302 | HIV+ |
| Woldetsadik AB [45] | 2020 | AA | 12.2 | FBCS | 425 | All |
| Aynalem BY et al. [54] | 2020 | Amhara | 5.4 | CBCS | 822 | All |
| Asres T [55] | Unpub | Amhara | 18 | FBCS | 322 | Healthcare |
| Dulla D et al. [59] | 2017 | SNNPR | 11.4 | FBCS | 367 | Healthcare |
| Heyi WD et al. [62] | 2018 | Oromia | 5.8 | CBCS | 845 | All |
| Berhanu T et al. [66] | 2019 | AA | 9.3 | CBCS | 291 | Healthcare |
| Tekle T et al. [60] | 2020 | SNNPR | 22.9 | CBCS | 520 | All |
| Ashagrie A [63] | Unpub | Oromia | 16 | FBCS | 318 | HIV+ |

AA: Addis Ababa; CSWs: Commercial sex workers.

CBCS: community based cross-sectional study; FBCS: facility based cross-sectional study.

region revealed that the highest cervical cancer screening utilization was observed in the SNNPR, 18.59 (95% CI: 9.65, 27.53) followed by Oromia region, 16.00% (95% CI: 16.00% (95% CI: 6.31, 25.7) and lowest occurred in Amhara region, 13.62% (95% CI: 9.92, 17.32) (Table 2). In addition, the pooled subgroup analysis showed that cervical cancer screening was highest in studies that were institution- based cross-sectional studies, 17.54% (95% CI: 13.16, 21.93). The highest cervical cancer screening was among HIV- positive women, 20.71% (95% CI: 12.8, 28.63) and the lowest was among reproductive age women, 11.54% (95% CI: 8.00, 15.05) (Table 2).

## Predictors of cervical cancer screening utilization

**Association of educational status and utilization of cervical cancer screening.** In regard to the social inequities, the effects of three predictors on cervical cancer screening utilization were estimated. Thus, age of women and occupational status were not significantly associated with cervical cancer screening utilization (S2 and S3 Figs). While, women's educational status was significantly associated with utilization of cervical cancer screening. Accordingly, the pooled random effect of eight studies [48–50, 54, 57, 62, 63, 60] found that women who have

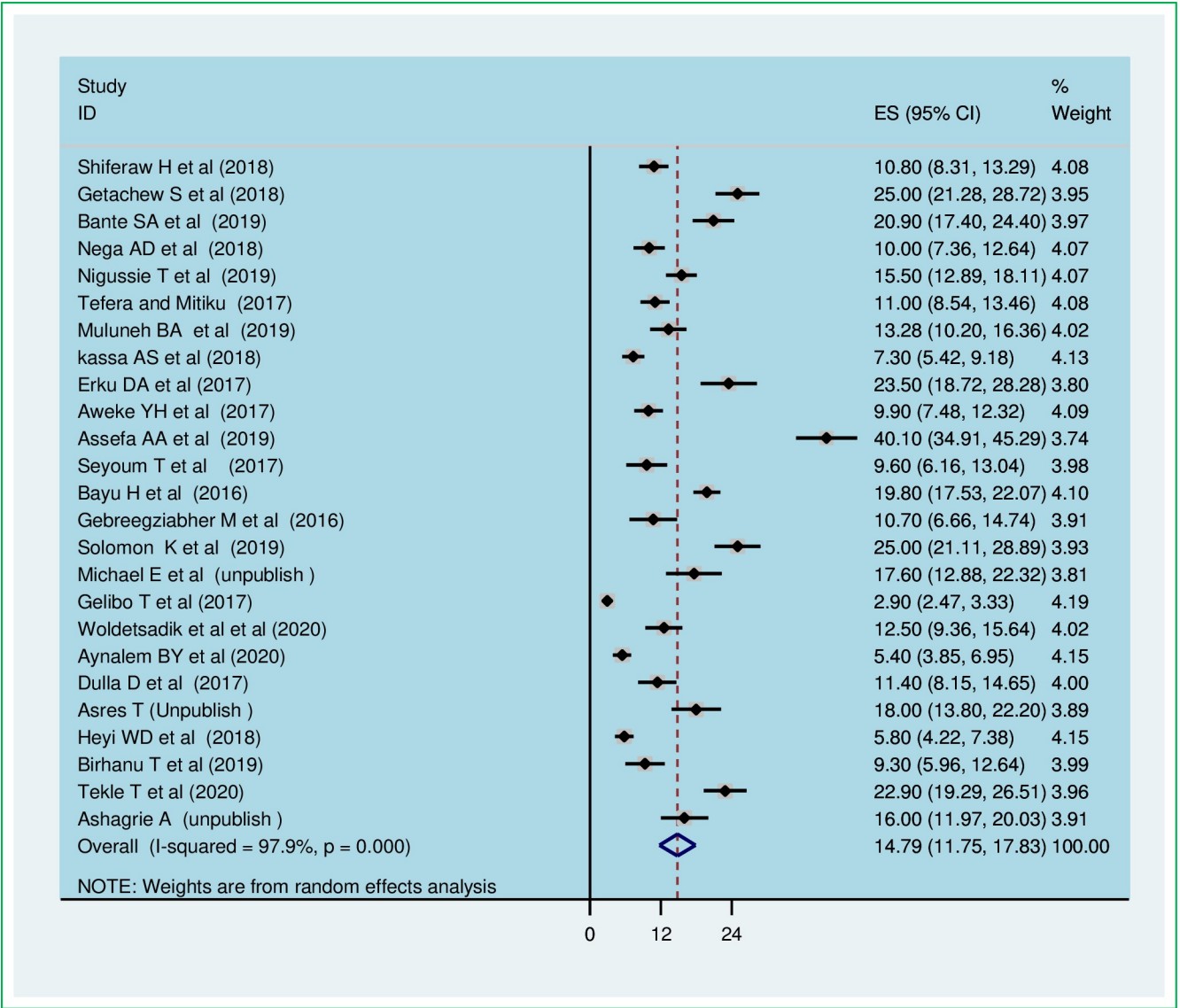

**Fig 2. The pooled utilization of cervical cancer screening among women in Ethiopia.**

no formal education were 66% (POR:0.33, 95% CI: 0.23,0.46) times less likely to utilize cervical cancer screening than those who attended any formal education (Fig 4).

**Association of knowledge and perception of cancer and screening utilization.** The meta-analysis of 14 studies revealed [42, 45, 49–51, 53, 54, 57, 58, 60, 62–65] that women's knowledge of cervical cancer screening uptake was the commonest predictor of screening utilization. Women who had good knowledge of cervical cancer screening reuptake were 3.97 times (POR: 3.49, 95% CI: 1.67, 7.33) more likely to have cervical cancer screening than women who had poor knowledge (Fig 5).

The pooled effect of six studies [33, 42, 45, 49, 53, 64] also revealed that the perceived susceptibility to cervical cancer was another major predictor of cervical cancer screening utilization in Ethiopia. Women who had perceived susceptibility to cervical cancer were 5.5 times more likely to reuptake cervical cancer screening than their counterparts (POR = 5.54, 95% CI:

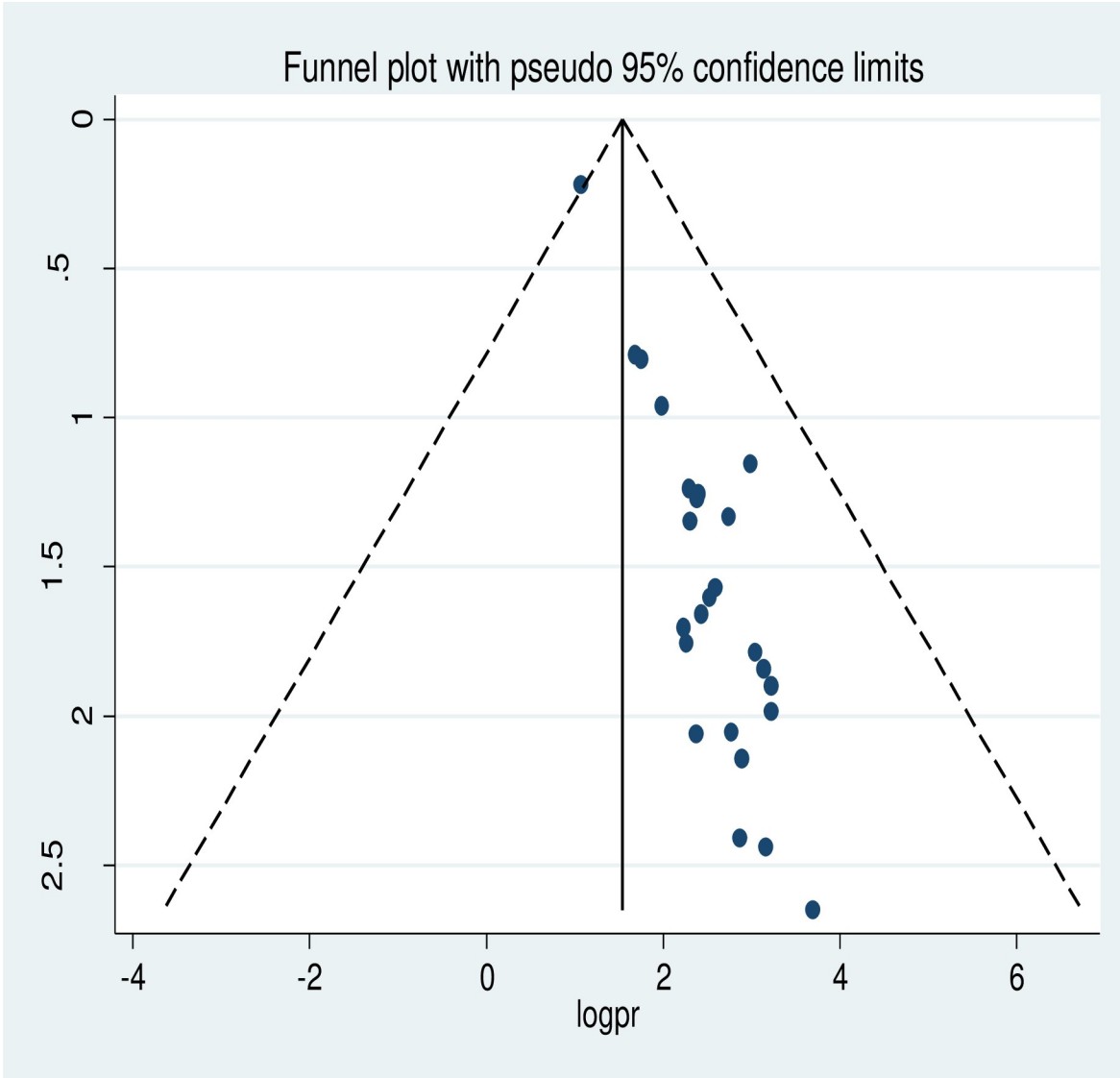

**Fig 3. Funnel plot of the prevalence of cervical cancer screening utilization in Ethiopia.**

4.28, 7.16) (Fig 6). Similarly, women who had perceived severity of cervical cancer were more likely to utilize cervical cancer screening (POR = 6.57, 95% CI: 3.99, 10.82) (Fig 7).

**Association of history of sexual transmitted infection and cervical cancer screening uptake.** Based on the pooled analysis of four studies [47, 51, 54, 64], women who had history of sexual transmitted infection were more likely to utilize cervical cancer screening (POR: 3.32, 95% CI: 1.07, 10.34) (Fig 8).

## Barriers of cervical cancer screening uptake

The pooled analysis also revealed that the most common reasons that hinder the use of cervical cancer screening were associated with women considered to be healthy, 48.97% (95% CI: 38.3, 59.59) and lack of information on screening, 34.34% (95% CI: (17.93, 50.75) (Table 3).

**Table 2. Sub-group analysis of cervical cancer screening utilization in Ethiopia: A meta-analysis.**

| Subgroup type | Category | No of studies | Prevalence(95%CI) | $I^2$ | P-value |
|---|---|---|---|---|---|
| **Study design** | FBCS | 12 | 17.54 (13.16,21.93) | 94.6% | <0.0001 |
| | CBCS | 13 | 12.29 (8.70,15.88) | 98.0% | <0.0001 |
| **Region** | Addis Ababa | 4 | 14.32 (8.09,20.56) | 93.7% | <0.0001 |
| | Amhara | 9 | 13.62 (9.92,17.32) | 94.5% | <0.0001 |
| | SNNPR | 5 | 18.59 (9.65,27.53) | 97.3% | <0.001 |
| | Oromia | 4 | 16.00 (6.31, 25.7) | 97.1% | <0.001 |
| | Tigray | 2 | 15.41 (6.5, 24.32) | 93.3% | <0.001 |
| | National level | 1 | 2.9 (2.47,3.33) | - | |
| **Women characteristics** | HIV positive | 5 | 20.71 (12.8,28.63) | 96.6% | <0.0001 |
| | All women | 12 | 11.54 (8.00, 15.05) | 97.9% | <0.0001 |
| | Healthcare workers | 4 | 12.21 (8.71,15.71) | 72.4% | 0.012 |
| | Commercial sex worker | 1 | 13.28 (10.2,16.36) | - | - |

## Discussions

The uptake of cervical cancer screening services in Ethiopia is not well established. Despite, WHO recommends cervical cancer screening tests to be included as part of well-planned and implemented programs in every country's health care policy. This systematic review and meta-analysis was conducted to estimate the pooled level of cervical cancer screening and its associated factors in Ethiopia. Accordingly, the pooled national level of cervical cancer screening utilization was 14.79 (95% CI: 11.75, 17.83). This was lower than 85% from a study conducted in United States [13], 21.4% in China national population based survey [67], 19.4% in Kenya [68], 19% - 63% from studies conducted in 54 countries [69], 48.9% in Malaysia [70], and also lower than 67% from a national-based study conducted among Vietnamese women [71]. The difference could be explained by the variation in the population characteristics, study settings and quality of health care services and screening programs. Besides, this could be explained by socio-economic inequalities, higher birth order and poor access to reproductive health care service utilization in Ethiopia could lower the cervical screening utilization. Previous report also showed that women with high birth order and poor women are less likely to receive cervical screening service [69]. In Ethiopia, a small proportion of women are in contact with obstetric or gynecological health services and that the health system may not have the capacity to provide effective screening to a larger number of women. Therefore, intervention programs to improve the quality of cervical cancer screening clinics are essential.

The findings of this meta-analysis also showed that the highest prevalence of cervical cancer screening occurred in the SNNPR followed by Oromia region and the lowest was in Amhara region. Regional variation in the burden of cervical cancer screening in Ethiopia might be explained by the difference in maternal health care service utilization that could be explained by in the difference in spousal support, cultural and linguistic diversity across the regions and societal stigmatization. Additionally, health service-related reason like cost of access to services, proximity to facilities, navigation of the facilities, waiting time and attitude of the health care staff may be the reasons for the regional difference and lower use of cervical cancer screening in the country.

The highest screening utilization in SNNPR and Oromia may be due to the nature of included studies in the respective regions. For example, 60% of the studies from SNNP region were institutional based cross-sectional studies and 50% of the included studies from Oromia region were conducted among HIV-positive women. Such differences may have contributed

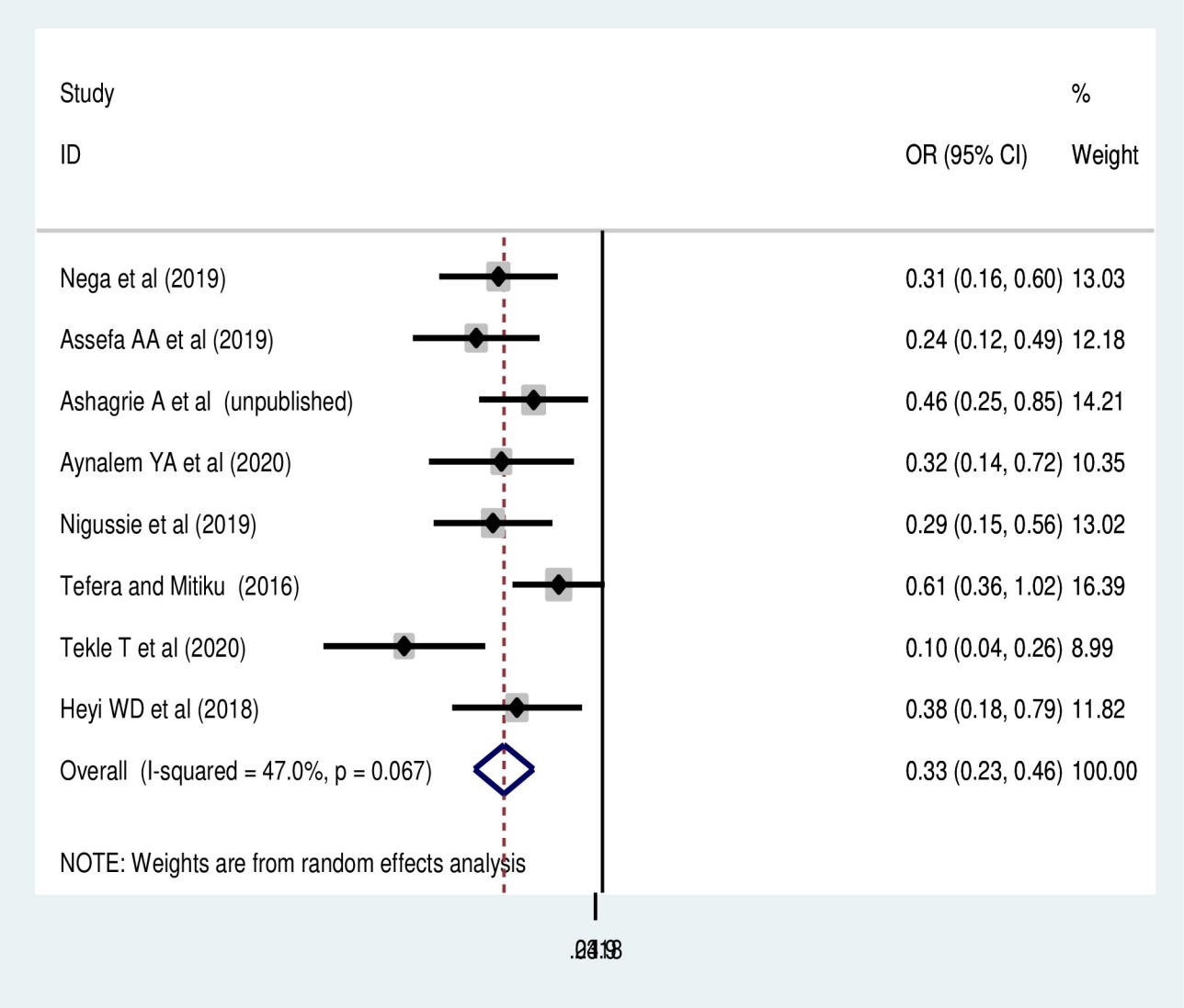

**Fig 4. Association of educational status with cervical cancer screening in Ethiopia.**

to the higher prevalence of cervical cancer screening in SNNPR and Oromia regions. Furthermore, socio-demographic characteristics and lifestyle activities could also be mentioned as reasons for the variation in screening across the different regions in the country. The pooled cervical cancer screening was also highest among HIV- positive women (20.71%). This may be due to the fact that these women may be given information about the disease during their follow-up visit to antiretroviral therapy [57], which may improve their knowledge about cervical cancer, and therefore, increase service utilization.

This systematic review and meta-analysis found that educational status of women was one of the significant predictors of cervical cancer screening utilization. No formal education reduces the cervical cancer screening uptake by 67%, and this finding was supported by a study done in China [67] and a meta-analysis conducted in developed countries [72]. The possible justification for this might be due to the fact that women who have no formal education are less likely to have gynecological examinations and maternal health service utilization. As

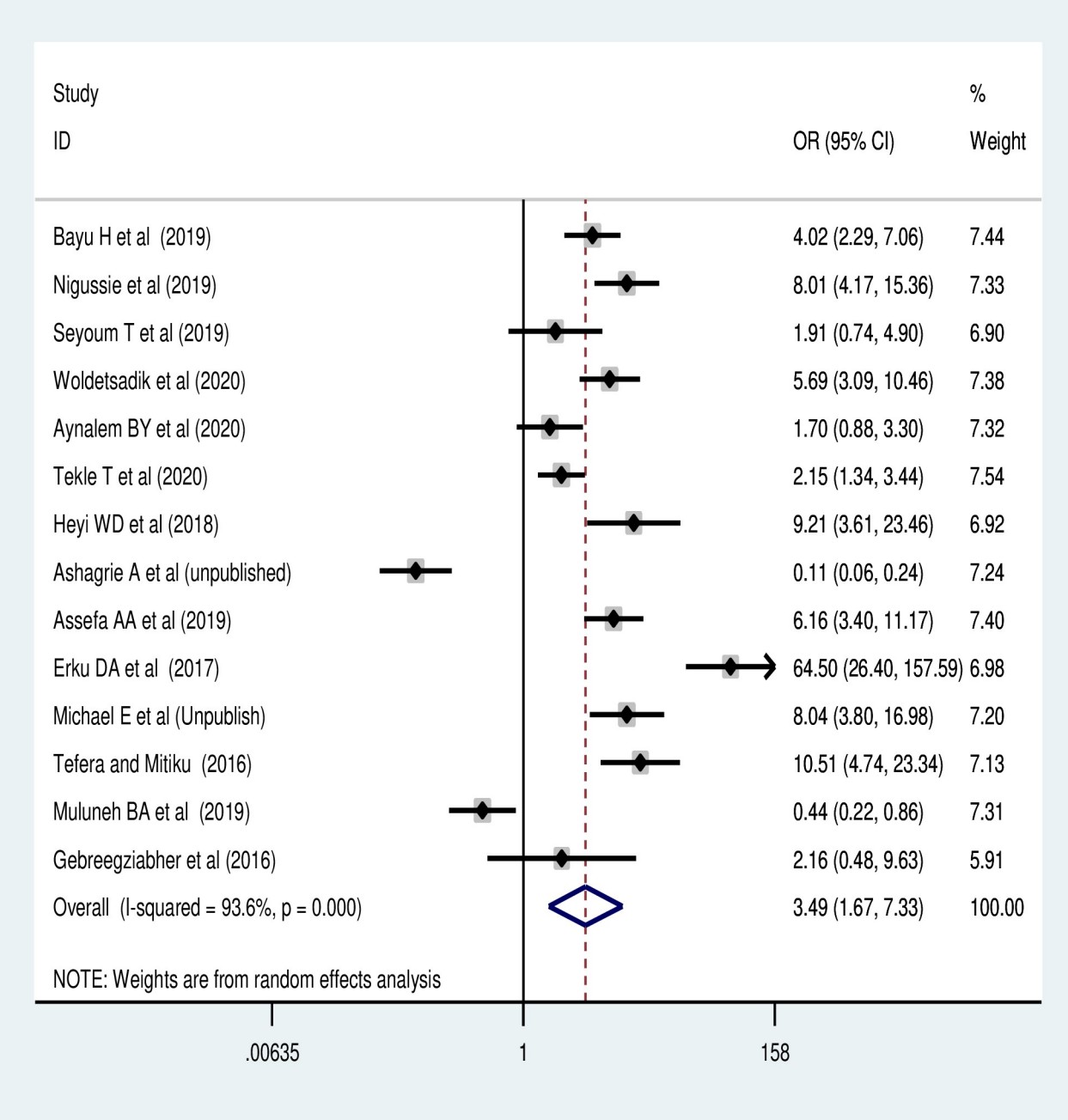

**Fig 5. Association of knowledge of the screening with cervical cancer screening utilization.**

the result, they are likely to have limited exposure to visit health institution for antenatal care, health facility delivery and post-natal care.

Uneducated women also have lower possibilities to read and fully understand the information and instructions provided by healthcare providers, and therefore, reduce the rate of cervical cancer screening. Cervical cancer educational interventions and provider recommendation for screening increases the rates of cervical cancer screening [73]. Therefore, more integrated interventions to improve women's empowerment should be done at national level to improve

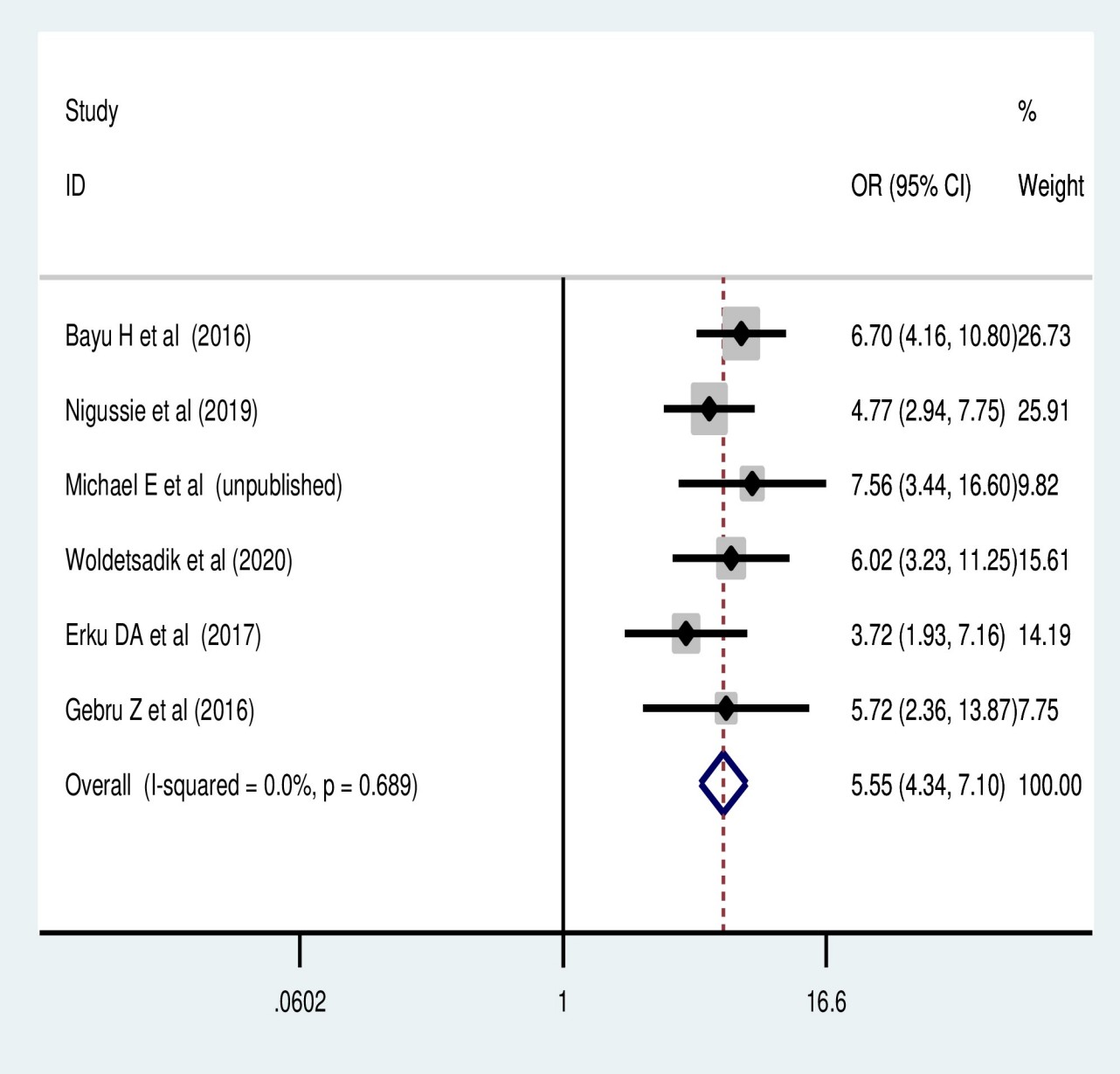

**Fig 6. Association of perceived susceptibility to cervical cancer with cervical cancer screening.**

the rate of cervical cancer screening utilization, and therefore, reduce cervical cancer related morbidity and mortality.

This study also found that women's knowledge of screening for cervical cancer was a significant predictor of cervical cancer screening service uptake. The finding was supported by studies done in Uganda [74], Malaysia [70], a review done in LMICs [75] and among Arab women [76]. This could be explained by the fact that those women who had good knowledge for cervical cancer screening are more likely to give priority to the issue and improves their decisions on health- seeking screening behavior. Accordingly, findings in Ethiopia, Malawi, Tanzania

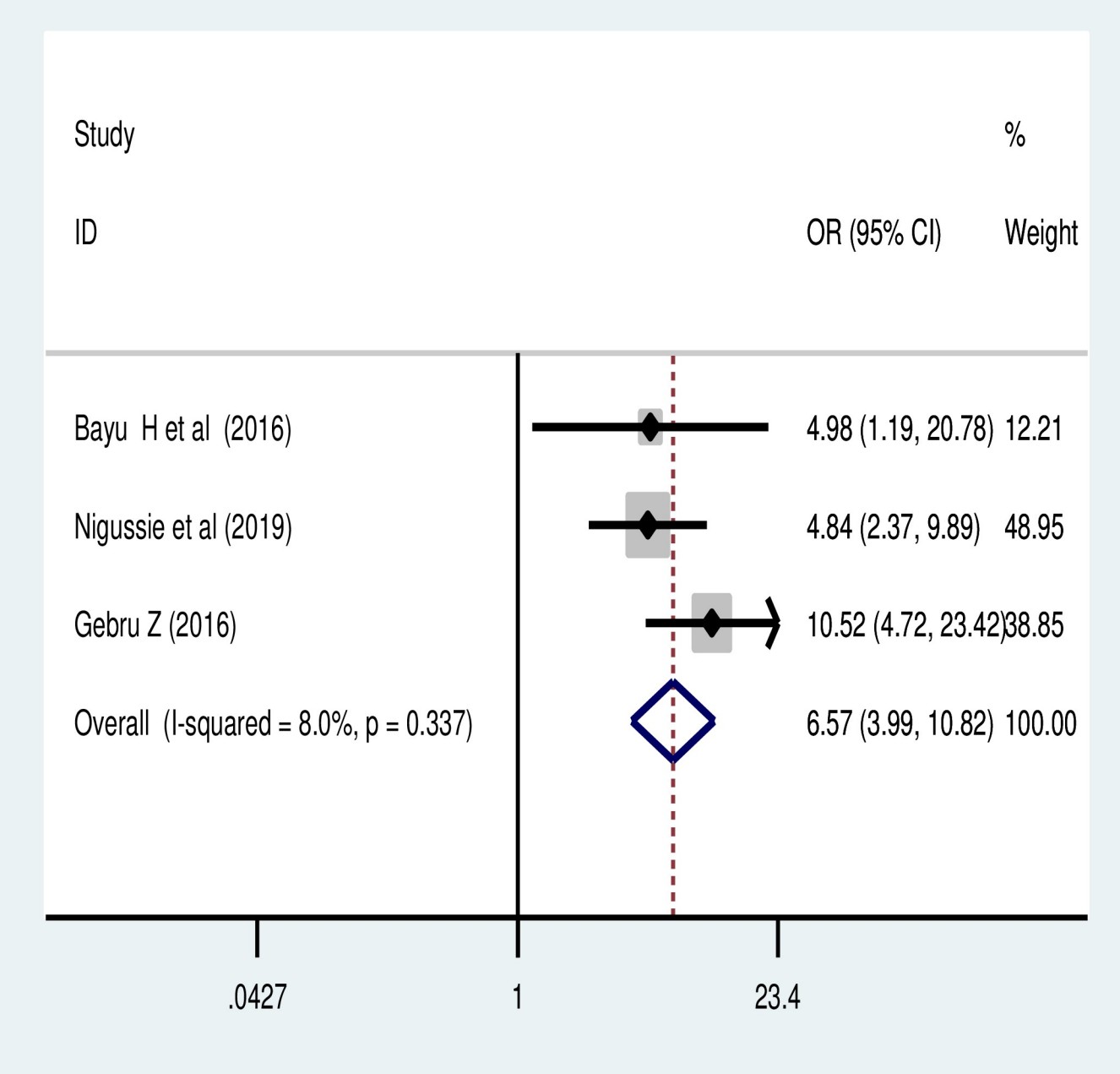

**Fig 7. Association of perceived severity of cancer and cervical cancer screening utilization.**

and Thailand [64, 77–79] have shown that a good flow of information and awareness creation campaigns about cervical cancer increase the uptake of cervical cancer screening.

This meta-analysis also showed that women with a history of STI were more likely to use screening for cervical cancer compared to those with no history of STI. This result was supported by the findings of other studies [64, 80]. This may be explained by the fact that women who have STIs and history of STI will have an increased chance of visiting health institutions for treatment and medical check-ups, and therefore, more likely to get the screening information from the healthcare provider.

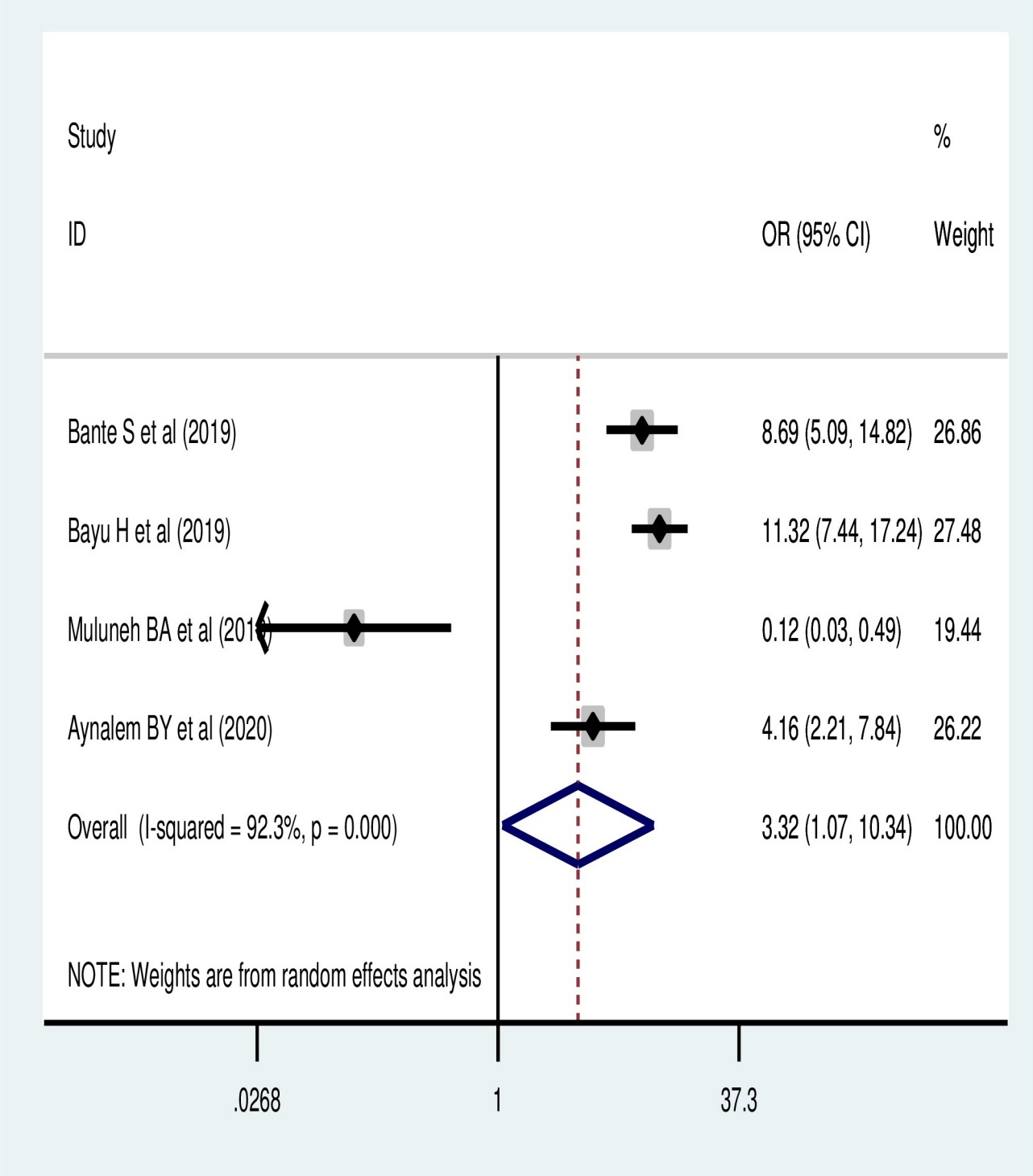

**Fig 8. Association of history of sexual transmitted infection with cervical cancer screening utilization.**

This systematic review and meta-analysis also found that perceived susceptibility and severity were also predictors of the use of cervical cancer screening as supported by Wanyenze et al. [74]. This may be those who perceive their susceptibility or severity of cancer may be aware of

**Table 3. Barriers of the cervical cancer screening utilization in Ethiopia: A meta-analysis.**

| Barriers | Studies | Prevalence [95% CI] | I² | P-value |
|---|---|---|---|---|
| Consider as healthy | 11 | 48.97% [38.3, 59.59] | 98.7% | <0.0001 |
| Fear of screening | 11 | 15.25% [6.77,23.73] | 99.4% | <0.0001 |
| Lack of information | 7 | 34.34% [17.93, 50.75] | 99.4% | <0.0001 |
| Embarrassment | 8 | 11.16% [5.76,16.56] | 99.3 | <0.0001 |
| Long waiting time | 7 | 21.58% [6.87,36.28] | 99.6 | <0.0001 |
| Don't know place | 5 | 10.06% [3.53,16.59] | 97.0 | <0.0001 |

the severity of the cancer and higher level of education about the disease as a result of the increased screening rate. As a result, those women who have an increased perception of susceptibility or severity of the disease may have higher education that has increased adherence to the cervical cancer screening [68]. These may include those women who are perceived to be more acutely aware of their risk, more interested and knowledgeable about health and behavioral issues, and better access to health information and resources [81]. This finding was also supported by recent studies done in Ghana [82] and Kenya [83] which found that women who perceived the severity of disease were more likely to accept screening due to increased perception of the benefits and barriers to cervical cancer, which increases their cancer screening.

Furthermore, the results of this systematic review and meta-analysis found that the common barriers to the utilization of cervical cancer screening were considered healthy and lack of information by women. This is supported by additional studies [47, 49, 51, 56, 64]. This may be due to the fact that those who consider their status to be healthy and who have poor knowledge are less likely to perceive the benefits of screening and the severity of cervical cancer., Therefore, multi-disciplinary interventions across the life course, community education and social mobilization on cervical cancer risk and its screening should be improved and emphasized to increase the cervical cancer screening utilization.

This review's strengths include the very extensive systematic search conducted and the inclusion of articles identified without specifying the population characteristics and period of publications. Our review adopted the international standard definitions to measure the quality of studies. This meta-analysis has its strengths because it has used a pre-specified protocol for search strategy and data abstraction and used internationally accepted tools for a critical appraisal system for the quality assessment of individual studies.

However, the results of this review should be interpreted with some limitation. The high heterogeneity in the characteristics of the studies might lead to insufficient statistical power to detect significant association. However, a meta-regression analysis revealed that there was no variation due to sample size and publication year. This meta-analysis was also unable to assess the type of screening, and therefore, an area of research for future studies. Additionally, the studies included in this review were from only five regions out of the nine regional states and the two administrative cities that might reduce its representativeness for the country. Some studies have small sample size, affect the estimation.

## Conclusions

This meta-analysis found that cervical cancer screening rate was lower than the WHO recommendations. Only one in every seven eligible women were screened in Ethiopia, and there was a significant variation in the screening level based on geographical regions and characteristics of women. Women's educational status, knowledge towards cervical cancer screening, perceived susceptibility and severity to cervical cancer and history of sexual transmitted infections

significantly increased uptake of the screening practice. Therefore, women empowerment, improving knowledge towards cervical cancer screening, enhancing perceived susceptibility and severity to cervical cancer and identifying previous history of women are an essential strategy to increase utilization of cancer screening. Moreover, adoption of the better strategies and addressing the barriers of cervical cancer screening uptake mainly improving of the provision of adequate information on cervical cancer screening has a paramount importance to improve cervical cancer screening among reproductive age women.

## Supporting information

**S1 Fig. Sensitivity analysis of cervical cancer screening utilization.**
(TIF)

**S2 Fig. Association of women's age and cervical cancer screening utilization.**
(TIF)

**S3 Fig. Association of occupational status and cervical cancer screening utilization.**
(TIF)

**S1 Table. PRISMA check list.**
(DOC)

**S2 Table. Included studies for the barriers of cervical cancer screening.**
(DOCX)

**S3 Table. Quality assessment of included studies.**
(DOCX)

**S1 Appendix. Specific searching on PubMed database.**
(DOCX)

## Author Contributions

**Conceptualization:** Melaku Desta, Yordanos Gizachew Yeshitila.

**Data curation:** Melaku Desta, Tewodros Eshete, Yordanos Gizachew Yeshitila.

**Formal analysis:** Melaku Desta.

**Funding acquisition:** Yordanos Gizachew Yeshitila.

**Investigation:** Melaku Desta, Tewodros Eshete.

**Methodology:** Melaku Desta, Tewodros Eshete, Getachew Mullu Kassa, Fentahun Adane, Yordanos Gizachew Yeshitila.

**Project administration:** Melaku Desta, Temesgen Getaneh.

**Resources:** Melaku Desta, Yordanos Gizachew Yeshitila.

**Software:** Melaku Desta.

**Supervision:** Temesgen Getaneh, Bewuket Yeserah, Yichalem Worku, Tewodros Eshete, Molla Yigzaw Birhanu, Getachew Mullu Kassa, Fentahun Adane, Yordanos Gizachew Yeshitila.

**Validation:** Melaku Desta, Bewuket Yeserah, Tewodros Eshete, Yordanos Gizachew Yeshitila.

**Visualization:** Temesgen Getaneh, Bewuket Yeserah, Yichalem Worku, Tewodros Eshete, Molla Yigzaw Birhanu, Getachew Mullu Kassa, Fentahun Adane.

**Writing – original draft:** Melaku Desta.

**Writing – review & editing:** Temesgen Getaneh, Bewuket Yeserah, Yichalem Worku, Tewodros Eshete, Molla Yigzaw Birhanu, Getachew Mullu Kassa, Fentahun Adane, Yordanos Gizachew Yeshitila.

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
