## [Decision Letter · Decision Letter 0]

16 Dec 2020

PONE-D-20-22948

Cervical cancer screening utilization and predictors among reproductive-age women in Ethiopia: a systematic review and meta-analysis

PLOS ONE

Dear Dr. Desta,

Thank you for submitting your manuscript to PLOS ONE. After careful consideration, we feel that it has merit but does not fully meet PLOS ONE’s publication criteria as it currently stands. Therefore, we invite you to submit a revised version of the manuscript that addresses the points raised during the review process.

Two reviewers have evaluated your manuscript and are largely positive about the study. However, they have asked for further details regarding the the choice of age range as well as outlining the limitations of the study and it's generalizability for the study areas. 

We look forward to receiving your revised manuscript.

Kind regards,

Nicola Stead

Senior Editor

PLOS ONE

Journal Requirements:

Reviewers' comments:

Reviewer's Responses to Questions

**Comments to the Author**

1. Is the manuscript technically sound, and do the data support the conclusions?

Reviewer #1: Yes

Reviewer #2: Yes

2. Has the statistical analysis been performed appropriately and rigorously? 

Reviewer #1: Yes

Reviewer #2: Yes

3. Have the authors made all data underlying the findings in their manuscript fully available?

Reviewer #1: Yes

Reviewer #2: Yes

4. Is the manuscript presented in an intelligible fashion and written in standard English?

Reviewer #1: Yes

Reviewer #2: Yes

5. Review Comments to the Author

Reviewer #1: General comment: Very good and useful review that could provide evidence on barrier to CC screening in Ethiopia

Specific comments

IN the abstract as well as under background section, it is stated “Despite a remarkable progress in the reduction of maternal mortality, cervical cancer is the 73 second most commonly diagnosed cancer and the leading cause of cancer related death among 74 African women”. I agree with the reduction of MM. However, in view of the fact that CC is a major public health challenge and with its continued pressure, is it not naïve to argue about reduction of maternal mortality without showing how CC screening has contributed to that or contrary to this maintained MM in consequence higher.

Mention is made “The incidence, death rate and morbidities associated with cervical cancer is 79 significantly varies across the world; higher in the developing nations than compared to the 80 developed countries”(4), while again it is stated “The prevalence of cervical cancer screening is much higher at the Western countries than SSA” (11, 12); The two statements looks contradictory

Method

There is no clarity on whether this review is ‘systematic’ or ‘met-analysis’ or ‘both’. If it is both, it is imperative to clarify which part of it is systematic and meta-analysis OR clarify why both were considered.

From the statement “All published and unpublished studies through April 7, 2020” there were quite few concerns. Firstly what was the lower time frame. Could it be any document from time immemorial to April 7, 2020? Secondly, how were the unpublished reports captured? Were there any criterion set to identify those?

While WHO’s recommended age of CC screening is 30-50, it is not clear why in this study age 18-49 years was chosen? Is it not contradictory?

Results

The studies were drawn from five geographic regions: Addis Ababa, Amhara, Southern Nations, Nationalities and Peoples, Oromia and Tigray with only one from the national- level study. Couple of concern here. Firstly, how was the 18,067 women considered to estimate the pooled… proportionated in light of population size. Given the fact that studies are not proportional to the population of the regions, it would be difficult to reach conclusion as desired – this is about comparability of results. Secondly, I am wondering if this study could mirror realities for the country at large. Perhaps clarifying those and stating limitations may help.

I am not convinced that what are stated as ‘most common reasons that hinder the use of cervical cancer screening’ in the result section and conclusion section are aligned. This needs to be checked against the finding and corrected

Discussion

The fact that it was not clear on what of this review is meta and systematic, the discussion suffers much missing what is being discussed. Perhaps with clarification on how the two contributed to this review may

Reviewer #2: Generally a good paper, congratulations, however, the work needs some grammatic revision for clarity. I have listed some below but kindly revise the entire paper checking for grammatic and punctuation errors.

Line 74- Update the reference and use more recent data than 2013

Line 77- Be consistent in presenting figures i.e 311000 should have a comma as with other figures

Line 78- Correct the grammar of the sentence which start as "The incidence, death ......"

Line 85-6- Correct the grammar of the sentence

Line 97-98- Correct grammar

Line 99-100- Correct grammar

Line 103- Correct the punctuation

Line 292-295- Sentence needs revision for clarity, attend to the grammar

Line 295-298- Sentence needs some revision for clarity

Line 307 - Clarify what "tall reproductive age" refers, sentence may need some revisions

Line 309- Add reference of relevant papers to support your interpretation

Line 367- Replace "representative" with representativeness

6. PLOS authors have the option to publish the peer review history of their article (what does this mean?). If published, this will include your full peer review and any attached files.

Reviewer #1: **Yes: **Mirgissa Kaba

Reviewer #2: **Yes: **Dr. Oscar Tapera

---

## [Author Response · Author response to Decision Letter 0]

8 Apr 2021

Dear editors and reviewer of Plos One

We would like to extend our deepest appreciation for devoting your time to review our manuscript entitled “cervical cancer screening utilization and predictors among eligible women in Ethiopia: a systematic review and meta- analysis”. Cervical cancer is the second most commonly diagnosed cancer and the leading cause of cancer death in African women. In 2013, there were approximately 236,000 deaths from cervical cancer worldwide and it is the most common cancer in east and middle Africa. Cervical cancer screening is an important intervention to redcue cervical cancer and its associated maternal mortality. Even though, the utilization of the screening is inconsistent across the country and affected by different barrier. Therefore, this systematic review and meta-analysis estimates the pooled utilization of cervical cancer screening and its predictors in Ethiopia. Overall the main finding of this review is means of an intervention based on the pooled cervical cancer screening utilization and its predictors, which might be used to improve maternal adverse outcomes in Low and middle income countries, subsequently means of achieving SDGs. 

Dear reviewer, there has been a major revision of the whole structure of the manuscript (Abstract, introduction, methods, results, discussion and conclusions) mainly with a correction of grammar errors. The further details regarding the choice of age range as well as outlining the limitations of the study and it's generalizability for the study areas are addressed. The reproductive group is substituted by eligible women as the global and who recommendations. We hope now the manuscript is clear and more acceptable than its previous version. We tried to state the limitations and generalizability issues in the limitation and strength section of the discussion. We have tried to present the response for each reviewer according to your comment what to suppose to do so. For this, here we have given our responses to each of the concerns you raised, highlighted by red color. Again, we would like to remind our strongest gratitude for your effort for the improvement of this manuscript and the response for each the points were addressed in the response to reviewers’ section. For this, I kindly request you to consider the paper for publication. Again, we would like to remind our strongest gratitude for your effort for the improvement of this manuscript.

 Regards 

Reviewer #1 

1. Abstract 

1.1. In the abstract as well as under background section, it is stated “Despite a remarkable progress in the reduction of maternal mortality, cervical cancer is the 73 second most commonly diagnosed cancer and the leading cause of cancer related death among 74 African women”. I agree with the reduction of MM. However, in view of the fact that CC is a major public health challenge and with its continued pressure, is it not naïve to argue about reduction of maternal mortality without showing how CC screening has contributed to that or contrary to this maintained MM in consequence higher. 

Response: thank you for the highly valuable scholarly comments and suggestions. Revision was made on the abstract and background section stating that maternal mortality reduction achieved with cervical cancer screening. 

1.2. Mention is made “The incidence, death rate and morbidities associated with cervical cancer is significantly varies across the world; higher in the developing nations than compared to the 80 developed countries”. While again it is stated “The prevalence of cervical cancer screening is much higher at the Western countries than SSA” (11, 12); the two statements looks contradictory.

 Response: Thank you for the comments, but, the two statements is non-contradictory. Hence, the incidence, death rate and morbidities associated with cervical cancer is significantly varies across the world, which is higher in developing countries like SSA. This is explained due to high prevalence and severity of the problem among in low income countries and the fact that it is the only gynaecologic cancer which can be prevented and treated through early screening and follow-up, and the cervical cancer screening practice in low income countries is significantly low. Therefore, the two statements support each other. 

#2. Method

2.1. There is no clarity on whether this review is ‘systematic’ or ‘met-analysis’ or ‘both’. If it is both, it is imperative to clarify which part of it is systematic and meta-analysis OR clarify why both were considered. 

 Response: Thank you for the valuable comments. We used the PRISMA diagram or recommendations of the systematic review and meta-analysis including the report of this study like the search strategy and the overall written was based on the recommendations. The meta-analysis commands and software were used based on the meta-analysis Cochrane handbooks. The meta-analysis was done to get the pooled estimate using Stata 14 using a meta-analysis written command. To do a good review and meta-analysis it should be searched systematically and other quality measures should be addressed what we have tried.

2.2. From the statement “All published and unpublished studies through April 7, 2020” there were quite few concerns. Firstly what was the lower time frame? Could it be any document from time immemorial to April 7, 2020? Secondly, how were the unpublished reports captured? Was there any criterion set to identify those? 

 Response: accepted and specific lower time frame was putted (2016) and the criteria used to identify unpublished articles were those fully access during manual search, found in university repositories and those studies fulfilling quality score to be included in this study. 

2.3. While WHO’s recommended age of CC screening is 30-50, it is not clear why in this study age 18-49 years was chosen? Is it not contradictory?

Response: Thank you for the highly valuable comments. I completely agree what you supposed to do so and revision was made throughout the manuscript. The reproductive age group is not representative for all the all age eligible women for cervical cancer screening for the population charactertics we included. The recommendation was for the general population Evidence show success of cervical screening initiatives depend on high participation of the target population, which in turn is determined by the women’s knowledge, perceptions, health orientations and other socio-cultural issues. It is also affected by factors including early marriage, early sexual practice, delivery of the first baby before the age of 20, too many or too frequent childbirths, multiple sexual partners and low socio economic status. Women with early sexual practice, multiple partners, having HIV AIDS and sexual transmitted disease should have more screening schedule. 

According to world health organization (WHO) guideline, every sexually active woman aged 30–49 years should undergo cervical cancer screening at least every 5 years. However, sexually active and HIV-positive women are suggested to be screened every 3 years regardless of their age. HIV positive women Ethiopia adopted the WHO recommendation in 2015 and recommended HIV positive women to start screening at HIV diagnosis, regardless of age once the woman is sexually exposed. In this meta-analysis age all age eligible women or from different population charactertics for cervical cancer screening such as women from 30-50 years old, HIV positive women, healthcare workers, and commercial sex workers were included. Thus, eligible women are better than reproductive age and we have accepted your comment. 

#3. Results

3.1. The studies were drawn from five geographic regions: Addis Ababa, Amhara, Southern Nations, Nationalities and Peoples, Oromia and Tigray with only one from the national- level study. Couple of concern here. Firstly, how was the 18,067 women considered to estimate the pooled… proportionated in light of population size. Given the fact that studies are not proportional to the population of the regions, it would be difficult to reach conclusion as desired – this is about comparability of results. Secondly, I am wondering if this study could mirror realities for the country at large. Perhaps clarifying those and stating limitations may help.

Response: accepted and revision was made. The limitations were stated in regard to the less representativeness of five studies out of nine regions in Ethiopia in our meta-analysis. 

3.2. I am not convinced that what are stated as ‘most common reasons that hinder the use of cervical cancer screening’ in the result section and conclusion section are aligned. This needs to be checked against the finding and corrected 

Response: thank you for the comments suggest being. 

We have checked it and the barriers in the result section and conclusion section are aligned

#4. Discussion

4.1. The fact that it was not clear on what of this review is Meta and systematic, the discussion suffers much missing what is being discussed. Perhaps with clarification on how the two contributed to this review may help refine the discussion section. 

Response: Highly valuable comment, I completely agree what you supposed and revision was made. In the last two paragraphs of the discussion section narrates about the meta-analysis and systematic review mainly the strength and weakness. 

Reviewer #2 

1. Generally a good paper, congratulations, however, the work needs some grammatical revision for clarity. I have listed some below but kindly revise the entire paper checking for grammatical and punctuation errors.

Response: Thank you for your scholarly comments.

The major grammatical errors were revised and seen by sinor experts. 

2. Line 74- update the reference and use more recent data than 2013

Response: Accepted and data published in 2016 by WHO and Bruni et al 2017 was cited. 

3. Line 77- Be consistent in presenting figures i.e. 311000 should have a comma as with other figures

4. Response: accepted and revision was made. Comma as with other figures is putted. 

5. Line 78- Correct the grammar of the sentence which starts as "The incidence, death ......"

Response: accepted and the grammar error was revised. 

6. Line 85-6- Correct the grammar of the sentence, Line 97-98- Correct grammar, Line 99-100- Correct grammar, Line 103- Correct the punctuation, Line 292-295- Sentence needs revision for clarity, attend to the grammar, Line 295-298- Sentence needs some revision for clarity.

Response: all grammar and punctuation errors was corrected 

7. Line 307 - Clarify what "tall reproductive age" refers, sentence may need some revisions

Response: valuable comment and revised. The editing "tall reproductive age" is an error, we want to instead want to spell as all reproductive age and revision was made. Fortunately, hence, it was not clear for reader we remove from the document. 

8. Line 309- Add reference of relevant papers to support your interpretation

Response: reference number 73 was cited. 

9. Line 367- Replace "representative" with representativeness

10. Response: representative" was replaced by with representativeness.

---

## [Decision Letter · Decision Letter 1]

19 Oct 2021

Cervical cancer screening utilization and predictors among eligible women in Ethiopia: a systematic review and meta-analysis

PONE-D-20-22948R1

Dear Mr Desta,

We’re pleased to inform you that your manuscript has been judged scientifically suitable for publication and will be formally accepted for publication once it meets all outstanding technical requirements.

Kind regards,

Gizachew Tessema, PhD

Academic Editor

PLOS ONE

Additional Editor Comments.

I would suggest to shorten the description related to key words and search terms. Instead put the details search terms and key words presented in lines 158-172 in a supplementary appendix. Revise the statement in the methods section of the abstract. Indicate that databases were searched for peer-review articles whereas Google scholar was used to search grey literature

Reviewers' comments:

Reviewer's Responses to Questions

**Comments to the Author**

1. If the authors have adequately addressed your comments raised in a previous round of review and you feel that this manuscript is now acceptable for publication, you may indicate that here to bypass the “Comments to the Author” section, enter your conflict of interest statement in the “Confidential to Editor” section, and submit your "Accept" recommendation.

Reviewer #1: All comments have been addressed

Reviewer #2: (No Response)

2. Is the manuscript technically sound, and do the data support the conclusions?

Reviewer #1: Yes

Reviewer #2: Yes

3. Has the statistical analysis been performed appropriately and rigorously? 

Reviewer #1: Yes

Reviewer #2: Yes

4. Have the authors made all data underlying the findings in their manuscript fully available?

Reviewer #1: (No Response)

Reviewer #2: Yes

5. Is the manuscript presented in an intelligible fashion and written in standard English?

Reviewer #1: Yes

Reviewer #2: Yes

6. Review Comments to the Author

Reviewer #1: Authors revised the manuscript following the comments. They considered the comments useful which helped them to refine the manuscript

Reviewer #2: Great paper that contribute to knowledge in LMICs. Please see a few comments below:

-Some grammar and punctuation corrections are needful throughout the paper to aid clarity

-Line 74 Use more recent data e.g GLOBOCAN 2020

-Line 77 Be consistent in presentation of figures use the format 311,000

7. PLOS authors have the option to publish the peer review history of their article (what does this mean?). If published, this will include your full peer review and any attached files.

Reviewer #1: **Yes: **Mirgissa Kaba, School of Public Health, Addis Ababa University

Reviewer #2: **Yes: **Oscar Tapera

---

## [Editor Report · Acceptance letter]

26 Oct 2021

PONE-D-20-22948R1 

Cervical cancer screening utilization and predictors among eligible women in Ethiopia: a systematic review and meta-analysis 

Dear Dr. Desta:

I'm pleased to inform you that your manuscript has been deemed suitable for publication in PLOS ONE. Congratulations! Your manuscript is now with our production department. 

Kind regards, 

on behalf of

Dr. Gizachew Tessema 

Academic Editor

PLOS ONE